# A CNN-transformer fusion network for COVID-19 CXR image classification

**Kai Cao**[1], **Tao Deng**[2,3]*, **Chuanlin Zhang**[2], **Limeng Lu**[1], **Lin Li**[4]

**1** Key Laboratory of China's Ethnic Languages and Information Technology of Ministry of Education, Northwest Minzu University, Lanzhou, Gansu, China, **2** School of Mathematics and Computer Science, Northwest Minzu University, Lanzhou, Gansu, China, **3** Key Laboratory of Streaming Data Computing Technologies and Application, Northwest Minzu University, Lanzhou, Gansu, China, **4** School of Computing, University of Leeds, Leeds, United Kingdom

* dttom@lzu.edu.cn

**Data Availability Statement:** All the data in this study are from third party data. The datasets generated during and analyzed during the current study are available at the Guangzhou Women and Children Medical Centre dataset

## Abstract

The global health crisis due to the fast spread of coronavirus disease (Covid-19) has caused great danger to all aspects of healthcare, economy, and other aspects. The highly infectious and insidious nature of the new coronavirus greatly increases the difficulty of outbreak prevention and control. The early and rapid detection of Covid-19 is an effective way to reduce the spread of Covid-19. However, detecting Covid-19 accurately and quickly in large populations remains to be a major challenge worldwide. In this study, A CNN-transformer fusion framework is proposed for the automatic classification of pneumonia on chest X-ray. This framework includes two parts: data processing and image classification. The data processing stage is to eliminate the differences between data from different medical institutions so that they have the same storage format; in the image classification stage, we use a multi-branch network with a custom convolution module and a transformer module, including feature extraction, feature focus, and feature classification sub-networks. Feature extraction subnetworks extract the shallow features of the image and interact with the information through the convolution and transformer modules. Both the local and global features are extracted by the convolution module and transformer module of feature-focus subnetworks, and are classified by the feature classification subnetworks. The proposed network could decide whether or not a patient has pneumonia, and differentiate between Covid-19 and bacterial pneumonia. This network was implemented on the collected benchmark datasets and the result shows that accuracy, precision, recall, and F1 score are 97.09%, 97.16%, 96.93%, and 97.04%, respectively. Our network was compared with other researchers' proposed methods and achieved better results in terms of accuracy, precision, and F1 score, proving that it is superior for Covid-19 detection. With further improvements to this network, we hope that it will provide doctors with an effective tool for diagnosing Covid-19.

## 1. Introduction

Covid-19 is a pulmonary disease caused by severe acute respiratory syndrome coronavirus 2 (SARS-CoV-2) in 2019 and is highly infectious, mutagenic, and Covid-19 and its novel mutant

(URL: https://data.mendeley.com/datasets/rscbjbr9sj/2), the MIDRC-RICORD dataset (URL: https://wiki.cancerimagingarchive.net/pages/viewpage.action?pageId=70230281#7023028 1bcab02c187174a288dbcbf95d26179e8) and the COVIDx CXR dataset (URL: https://www.kaggle.com/datasets/andyczhao/covidx-cxr2). The authors did not have any special access privileges that others would not have.

**Funding:** This research was supported by the National Natural Science Foundation of China [China, 32060234], the Natural Science Foundation of Gansu Province [China, 21JR7RA164], the Fundamental Research Funds for the Central Universities [China, 31920210084] The funders had no role in study design, data collection and analysis, decision to publish, or preparation of the manuscript.

**Competing interests:** The authors have declared that no competing interests exist.

strains such as delta, omicron, and omicron XE variant have become pandemic worldwide [1,2]. On 30 January 2020, the World Health Organization (WHO) recognized the outbreak as a public health emergency of international concern (PHEIC) [3] and identified it as a pandemic on 11 March 2020 [4]. According to the WHO weekly epidemiological update on COVID-19 (12 April 2022), as of 10 April 2022, over 496 million confirmed cases and over 6 million deaths have been reported globally [5]. Therefore, detecting Covid-19 positive patients in the population at an early stage is not only important to curb the virus transmission and mutation, but also crucial to make disease staging and present treatment plans.

Currently, the main method for testing Covid-19 patients worldwide is Reverse transcription polymerase chain reaction (RT-PCR) [6]. RT-PCR is the gold standard for detecting viral Ribonucleic Acid (RNA), however, in some cases, the sensitivity of RT-PCR appears to be lower than computed tomography (CT), with 71% vs 98% according to the reports. The lower sensitivity is caused by the possible inadequate supply of reagents, the lack of expertise required for the testing, low viral load in patients, and the long testing cycles [7]. Unfortunately, if Covid-19 mutates during transmission, the epidemic will likely spread rapidly, with large numbers of cases appearing instantly. As a result, the task of rapidly detecting Covid-19 in a large population poses a great challenge to medical institutions worldwide.

Medical imaging and deep learning (DL) can play an important role in pre-detection efforts to combat disease. In recent years, researchers have used deep neural networks to achieve remarkable results in a variety of fields. Recent advances in DL show that computers can extract more information from images, more reliably, and more accurately than ever before [8,9]. However, further developing and optimizing DL techniques for the characteristics of medical images and medical data remains important but challenging research [10]. For example, ground-glass opacities are evident on chest X-ray or CT images for patients with Covid-19 [11,12]. Thus, a chest radiology-based system could be an effective way to detect, quantify and track Covid-19 cases. Furthermore, nature-inspired and heuristic optimization algorithms have been successfully adopted for various applications of medical images. For example, the use of a heuristic red fox heuristic optimization algorithm (RFOA) for medical image segmentation [13]. Nowadays, DL is increasingly applied to medical image classification, object detection, segmentation, and other tasks, and is replacing traditional machine learning methods in medical imaging [14].

Convolutional neural networks (CNN) [15–21] are one of the main research methods for solving computer vision tasks. CNN characterizes the images by abstracting local features at different levels. Automatic CNN-based Chest X-Ray (CXR) image classification for detecting Covid-19 attracted so much attention. Civit-Masot et al. used VGG16 to classify Covid-19 and achieved good results with an accuracy of 86% [22]. Ozturk et al. used a dark Covid-19 network for multiple classification experiments on Covid-19 with an accuracy of 87% [23]. Apostolopoulos et al. conducted training and testing using 224 Covid-19 images, 700 other pneumonia images, and 504 normal images for evaluating the classification performance of five pre-trained CNN networks, achieving the accuracy of 98.75% and 93.48% in two categories and three categories classification, respectively [24]. Yoo et al. used the ResNet18 model for Covid-19 detection with an accuracy of 95% [25]. Sethy et al. combined CNN and support vector machine (SVM) for Covid-19 classification, CNN for feature extraction, and SVM for classification, in which ResNet50 and SVM classifier got the best performance with an accuracy of 95.33% [26]. Minaee et al. obtained an accuracy rate of 95.45% using Squeeze Net [27]. Wang et al. constructed the deep CNN network called COVID-Net for the detection of Covid-19 cases from CXR images. In this study, a human-machine collaborative design strategy is adopted using human-driven principled network design prototyping and human-driven principled network design prototyping. The architecture adopts a lightweight projection-

expansion-projection-extension (PEPX) design, which can enhance the representation capability greatly and reduce the computational complexity, achieving better classification results [28]. Khan et al proposed a CNN-LSTM and improved max value features optimization framework to address the issue of multisource fusion and redundant features [29] and they also proposed a deep learning and explainable AI technique to select the best features for the diagnosis and classification of COVID-19 [30]. Arias-Garzón et al. proposed a new approach using existing DL models, which focuses on enhancing pre-processing stage to obtain accurate and reliable classification results. The pre-processing stage consists of a projection-based filtering network to divide the data into frontal or lateral, a segmentation model to extract lung regions containing relevant information, and a migration learning VGG classification model for classification with an accuracy of 97% [31]. Ahmed et al. studied four classification methods based on X-ray images and CT from three aspects: pre-processing, feature extraction, and classification, and proposed the use of Convid-Net to classify Covid-19 with an accuracy of 97.99% [32]. Islam et al. proposed a detection system for Covid-19 based on the combination of LSTM (Long Short-Term Memory) and CNN, where CNN was used for deep feature extraction and LSTM was used to classify the extracted features, with an accuracy of 99.4% [33].

Recently, the application of transformer [34–43] to the area of computer vision tasks increasingly demonstrates unique advantages. Vision transformer (ViT) uses a combination of a self-attention mechanism and a multi-layer perceptron (MLP), which reflects complex spatial transformations and long-range feature dependencies. Unlike that the CNN pays attention to local features, the transformer focuses on the global representation of images. Inspired by the transformer's success in natural language processing (NLP), Dosovitskiy et al applied a standard transformer directly to images, with the fewest possible modifications. To do so, an image is split into patches and the sequence of linear embeddings of these patches are provided as an input to a transformer. Image patches are treated the same way as tokens (words) in an NLP application [35].

For the detection of Covid-19 caused by SARS-CoV-2, this study proposes a classification network with CNN and transformer fusion, which automatically classifies the chest radiographs acquired during medical examinations. This network could assist doctors to judge whether a patient contracts pneumonia, furthermore to detect Covid-19 or bacterial pneumonia.

Data sets are collected from different medical institutions to enhance the applicability and robustness of this model. Data differences between medical institutions are reduced in the data processing stage, and a CNN- transformer fusion network is utilized for classification.

The main contributions of this work are as follows.

1. A fusion network of CNN and transformer is presented for COVID-19 CXR image classification.

2. Both local and global features are obtained and fed into two branches for feature extraction and finally fused for classification.

## 2. Method

Fig 1 overviews the two main stages of the proposed network for the automatic detection of pneumonia in X-ray images: data processing and image classification.

### Data processing

The data processing stage includes data transformation, data augmentation and adaptive normalization.

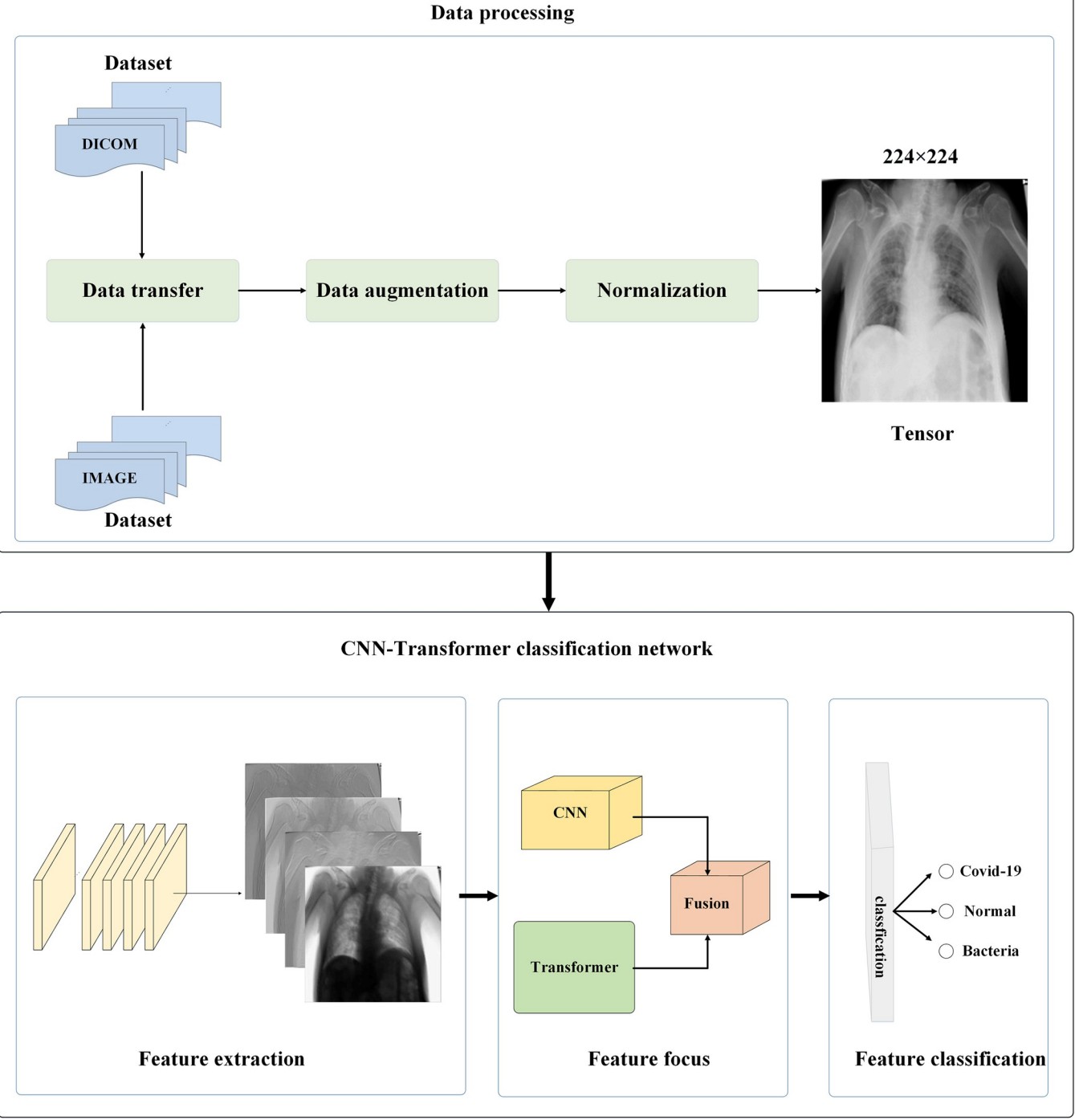

**Fig 1. Overview of image classification method based on CNN and transformer.**

## Data transformation

Datasets collected in this study come from different medical institutions, in image or DICOM file formats. Thus, the data will be converted to the tensor of the same resolution to ensure their uniformity.

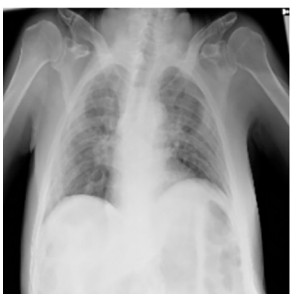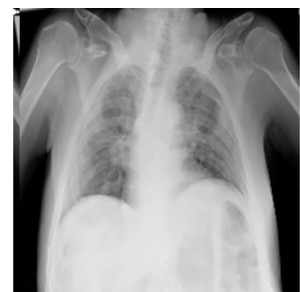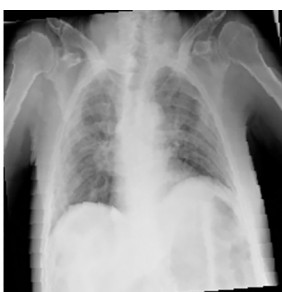

**Fig 2. Translation and rotation of the chest X-ray image.** (a) original image, (b) translated image, (c) Rotated image.

## Data augmentation

There are two problems regarding model universality in DL: the large amount of data required to train the model and the imbalance of data category. The number of training samples varies greatly from category to category, which causes problems in the learning process of the classification task. To address these issues, image translation and rotation are required in this work to augment the dataset and balance the category.

Image translation: The chest X-ray image will be randomly translated horizontally and vertically, $(\Delta x, \Delta y)$ is the amount of random translation and is determined by the image resolution.

Image rotation: The chest X-ray image will be rotated randomly clockwise or counterclockwise around the geometric center, $\theta$ is the angle of random rotation and is also determined by the image resolution.

Fig 2 shows the enhanced images, where (a) is the original image, (b) is the image of a horizontal translation of 5 pixels and (c) is the image of a counter-clockwise rotation of 5˚.

## Normalization

Medical images differ significantly from natural images in terms of dynamic range, natural images have a dynamic range of 3×255, while medical images can have a dynamic range of several thousand, such as CT, and even medical images may have a dynamic range of floating-point data, such as X-rays. Fig 3 shows the scattergram of (a) the medical image and (b) the natural image.

From Fig 3, the medical image differs greatly from the natural image in values. The maximum-minimum normalization is used in this work to normalize the radiological values in the

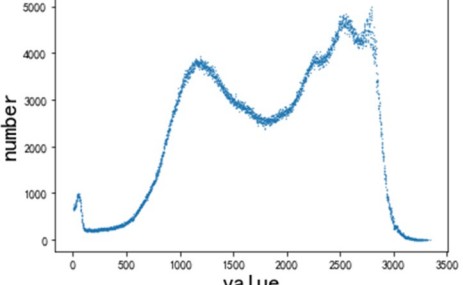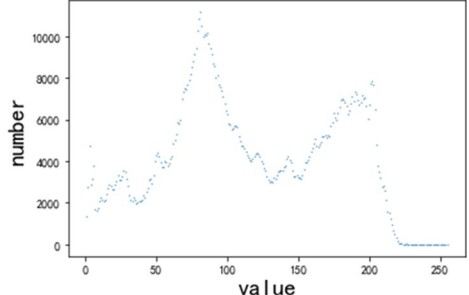

**Fig 3.** Statistics of values of (a) the medical image and (b) the natural image.

image $I$ to the interval $[a,b]$ according to Eq (1).

$$I_N = \frac{b-a}{I_{Max} - I_{Min}} \cdot I \qquad (1)$$

In Eq (1), $I$ is the original image, $I_{Max}$ and $I_{Min}$ are the maximum and minimum values extracted from $I$. In this study, $b$ and $a$ are specified as 1 and 0 respectively, and the normalized image $I_N$ is calculated by this equation.

## CNN and transformer network

**CNN module.** In DL, a CNN is a class of artificial neural network (ANN) commonly applied to image processing. CNN, thought to be shift-invariant and space-invariant, is based on the shared-weight architecture of the convolution kernels or filters that slide along input features and provide translation-equivariant responses known as feature maps. CNN uses multiple convolutional kernels at different levels to collect local features of images for representation and it has a unique advantage in extracting local features of images. CNN could enrich the extraction of hierarchical features and enhance its representation as the depth of the network increases. The residual structure in ResNet can effectively solve the network degradation as the depth of the network increases [18]. Fig 4(A) shows a basicblock, in which BacthNormal follows the down-sampled 3×3 spatial convolution and the 3×3 spatial convolution, and the identity mapping by shortcuts is between the basicblock input and the convolution output. The convolution module in this work contains L (L>1) basicblocks.

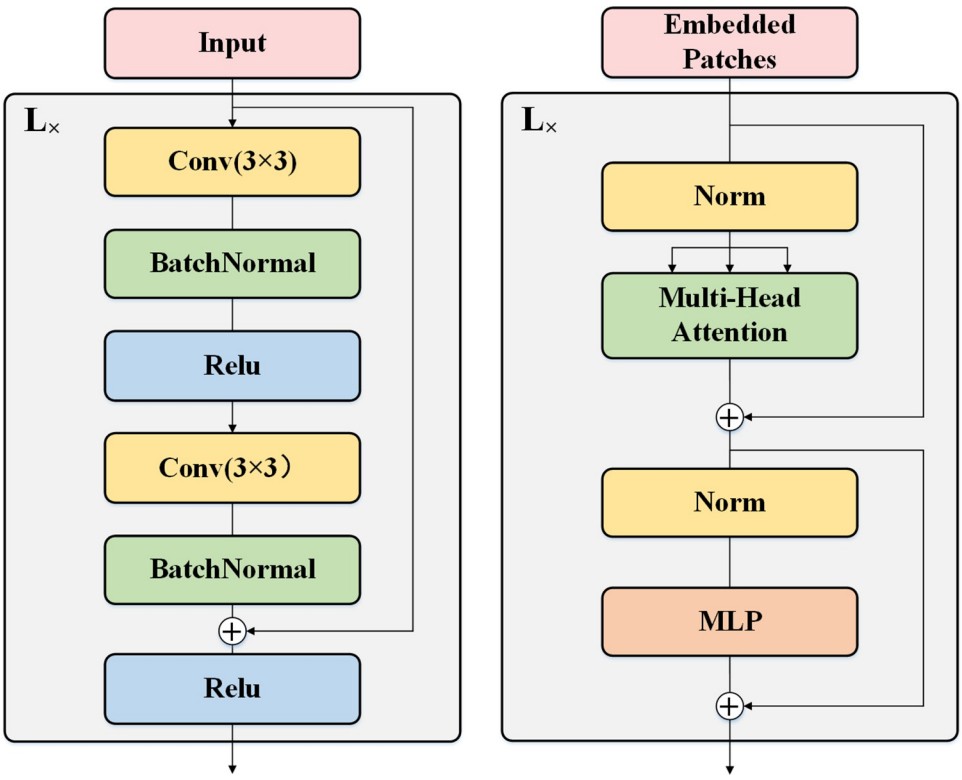

**(a) Basicblock module      (b) Transformer module**

**Fig 4. Convolutional modules and transformer modules.** (a) Basicblock module and (b) Transformer module.

**Transformer module.** Transformer, an architecture consisting of self-attention and MLP, uses the multi-head attention mechanism to obtain spatial transformations and long-range feature dependencies of an image to extract global features of the image.

Self-attention is the core of the transformer, which corresponds to the queue and a set of values to the input, forming a mapping of query $q$, key $k$, and value $v$ to the output. The output can be seen as a weighted sum of values, and the weight is derived from self-attention. Through the self-attention mechanism, the inputs of $x_1$ and $x_2$ are transformed into $z_1$ and $z_2$, and the equations are as follows.

$$\begin{aligned} q_1 &= x_1 W^Q & q_2 &= x_2 W^Q \\ k_1 &= x_1 W^K & k_2 &= x_2 W^K \\ v_1 &= x_1 W^V & v_2 &= x_2 W^V \end{aligned} \tag{2}$$

$W^Q$, $W^K$ and $W^V$ are three weight matrices. In the above equations, $x_1$ and $x_2$ share the same weight matrix $W$, and by this operation, information exchange is made between the vectors of $x_1$ and $x_2$.

$z_1$ and $z_2$ are obtained by the linear combination of $v_1$ and $v_2$, and $\theta$ is the combination of weights. The equation is as follows.

$$\begin{aligned} z_1 &= \theta_{11} v_1 + \theta_{12} v_2 \\ z_2 &= \theta_{21} v_1 + \theta_{22} v_2 \end{aligned} \tag{3}$$

$$[\theta_{11}, \theta_{12}] = \mathrm{softmax}\left( \frac{q_1 k_1^T}{\sqrt{d_k}}, \quad \frac{q_1 k_2^T}{\sqrt{d_k}} \right)$$

$$[\theta_{21}, \theta_{22}] = \mathrm{softmax}\left( \frac{q_2 k_1^T}{\sqrt{d_k}}, \quad \frac{q_2 k_2^T}{\sqrt{d_k}} \right) \tag{4}$$

$$\mathrm{Attention}\,(Q, K, V) = \mathrm{softmax}\left( \frac{Q K^T}{\sqrt{d_k}} \right) V \tag{5}$$

The encoder part of the transformer module for image classification tasks is often used in ViT, as shown in Fig 4(B). Each transformer encoder contains a multi-head self-attention module and an MLP module, and LayerNorm is before the multi-head self-attention module and the MLP module. The embedded patches input is connected to the transformer encoder using residuals.

## Proposed network

The features of medical images include obvious local lesion features and scattered global features. Thus, this study proposes a three-stage image classification network based on CNN-transformer, which consists of feature extraction, feature focus, and feature classification sub-networks. In this model, the local features of the image are extracted using the convolution module and the global features of the image are extracted by the transformer for fusion. This fusion could obtain the lesion features with both local and global features and get better classification results.

**Feature extraction sub-network.** The structure of the feature extraction sub-network is shown in Fig 5. The image tensor is convolved by a 5×5 convolution kernel with the stride of 2 and a 3×3 convolution kernel with the stride of 2. We measured the effect of different

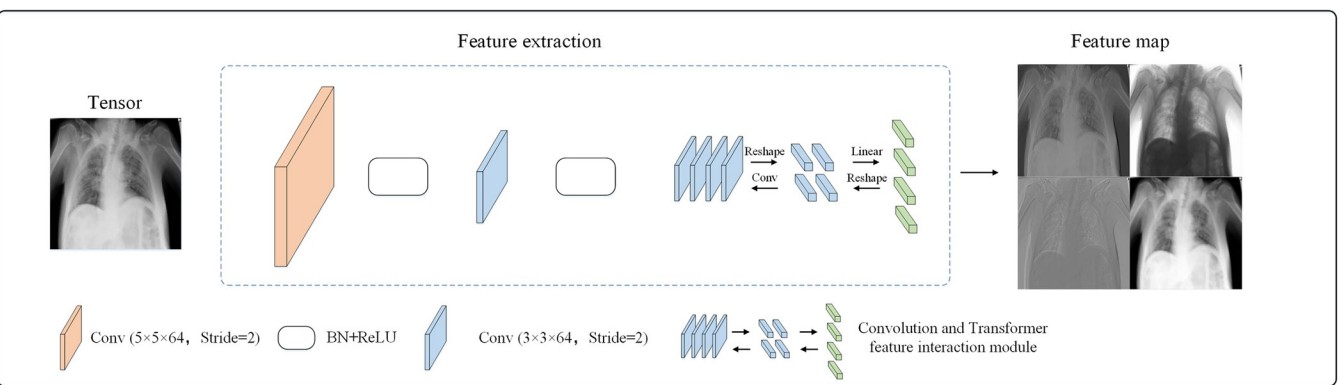

**Fig 5. Feature extraction sub-network.**

convolution fields in Section 4, and the extraction of local features with the convolution kernel is 5×5 is better. Batch normalization (BN) and rectified linear unit (ReLU) follow each convolution layer. However, the feature map extracted by CNN does not match the feature dimension of the transformer. In detail, the feature vector extracted by the CNN is H × W × C (H, W and C are height, width, and channel, respectively), while the encoded shape after the transformer is (N + 1) × D (N, 1, D are the number of patches, category labels, and output dimension, respectively). In this study, feature maps extracted by the convolution layer are converted into 7×7 patches by the customized convolution and transformer feature interaction module, then the downsample of patches through a linear layer is the same as embedding patch in dimension, and is added to embedding patch for feature fusion. The global features of the multi-head self-attention concerns are converted into a 56×56 tensor, and the tensor is projected using the 1×1 convolution layer and added to the tensor using 3×3 convolution layer for feature fusion. This makes it possible to realize information interaction of the local features with the global features.

**Feature focus sub-network.** The feature focus sub-network is made up of two modules, the CNN and the transformer. The CNN branch consists of 3 different basicblock modules, the first consisting of 2 with a step size of 1 and a convolutional kernel of 64, the second consisting of 2 with a step size of 2 and a convolutional kernel of 128, and the third consisting of 2 with a step size of 2 and a convolutional kernel of 256. The transformer branch consists of 8 transformer modules. As Fig 6 shows, the local features by the convolutional extraction are becoming more and more complex and abstract, while the global features are aggregated through the self-attention mechanism of the transformer.

**Feature classification sub-network.** Fig 7 shows the structure of the feature classification sub-network, where the features extracted from different modules of the feature focus sub-network are fused to obtain local and global features, and the final feature vectors are generated by the Average Pool and linear layers and are outputted through the softmax layer to predict the categories of Covid-19, normal and bacteria pneumonia.

## 3. Experiment and analysis

This section is concerned with the evaluation of the proposed model. To begin with, the data set and the parameters setting are specified to start the experiment. Next, the proposed model is compared with some DL-based models on this dataset, then with some other models regarding the detection of Covid-19.

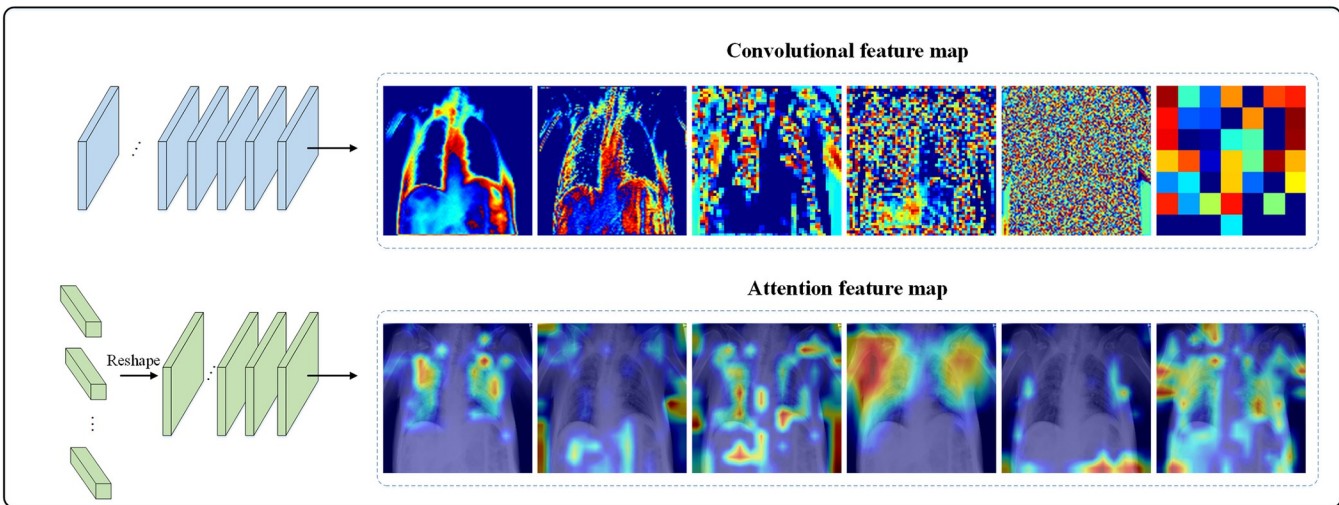

**Fig 6. Feature focus sub-network.**

## Dataset

The data in this study come from three medical institutions: Guangzhou Women and Children Medical Centre dataset [44], MIDRC-RICORD [45–47], COVIDx CXR dataset [28]. The categories of collected data are Covid-19, bacterial pneumonia, and normal. Data pre-processing was performed on the collected data and the distribution of the data after pre-processing is shown in Table 1.

## Experimental setup

Accuracy, Precision, Recall, and F1 score were used as the evaluation metrics. The experiments were carried out on the 64-bit operating system of Red Hat 4.8.5–28. The 4-card parallel training was conducted on Intel(R) Xeon(R) E5-2630 and Tesla M60 GPU, and each graphics card was executed on a server with a storage capacity of 8 GB memory. Under Pytorch version 1.9.1, CUDA 10.2 and CUDNN 7.6, the model was built and trained, with the training parameters in Table 2.

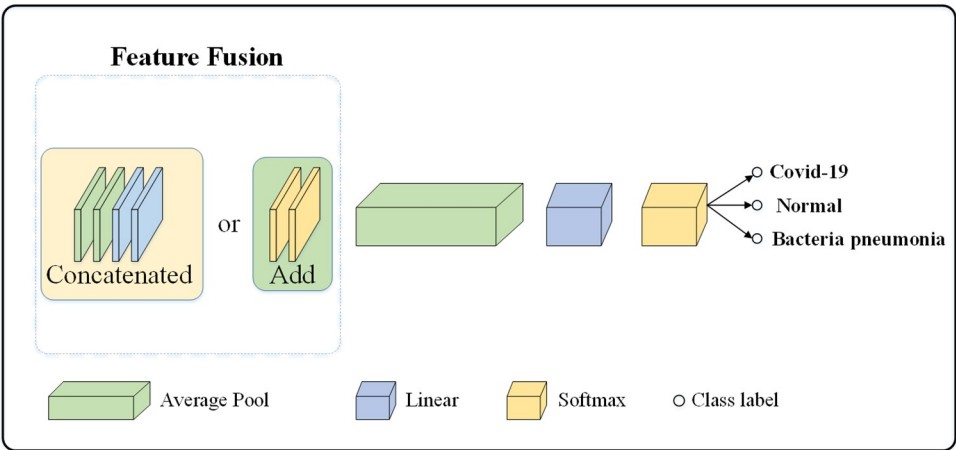

**Fig 7. Feature classification sub-network.**

**Table 1. The detailed description of the datasets used in this work.**

| Type | Normal | Bacteria pneumonia | Covid-19 | Total |
|---|---|---|---|---|
| train | 9,437 | 10,659 | 12,115 | 32,211 |
| validation | 3,146 | 3,553 | 4,038 | 10,737 |
| test | 3,146 | 3,553 | 4,039 | 10,738 |

## Experimental result

Fig 8 shows the confusion matrix derived from the experimental classification results. The proposed model is compared with four convolutional models and ViT, and Table 3 shows the evaluation metrics of the different models on the present dataset. Table 4 illustrates the proposed model in comparison to some other models regarding the detection of Covid-19.

The results show that the proposed model is more suitable than other models in classifying Covid-19 images. This may be due to that local and global features of the lesion are equally important for the diagnosis of Covid-19. In the previous study, researchers tend to focus more on local features and realize lesion classification by aggregating local features. The network proposed in this study focuses both on local lesion features and scattered global features in Covid-19 images. The fusion of two features solves the problem caused by paying more attention to local features than global features in characterizing lesions, thus achieving better results.

## 4. Discussion

This section compares the effects of possible module combinations, different convolutional kernel sizes, and mutual fusion and one-way fusion as well.

### Possible module combinations

As well known that local features become progressively more abstract as the convolution layer gets deeper, and global features become progressively more decreasing as the transformer is extracted. Our proposed network can extract different features and fuse features from different branches through mutual feature fusion to reduce the loss of useful feature information. Discussing the fusion of the local features with the global features, we conducted some experiments to test possible module combinations. As shown in Table 5, the fusion of the Transformer_block1 and the Conv_block1 gains the best results.

### Different convolution kernel size

CNN acquires local features through convolutional kernels, by which different feature maps are outputted. Feature extraction sub-network extracts local features and the transformer extracts global features for fusion, achieving good experimental results on the experimental

**Table 2. Parameters setting of the proposed classification network.**

| Parameter | Value |
|---|---|
| Learning rate | 0.001 |
| Gamma | 0.7 |
| Optimizer | Adam |
| Batch size | 32 |
| Epoch | 200 |

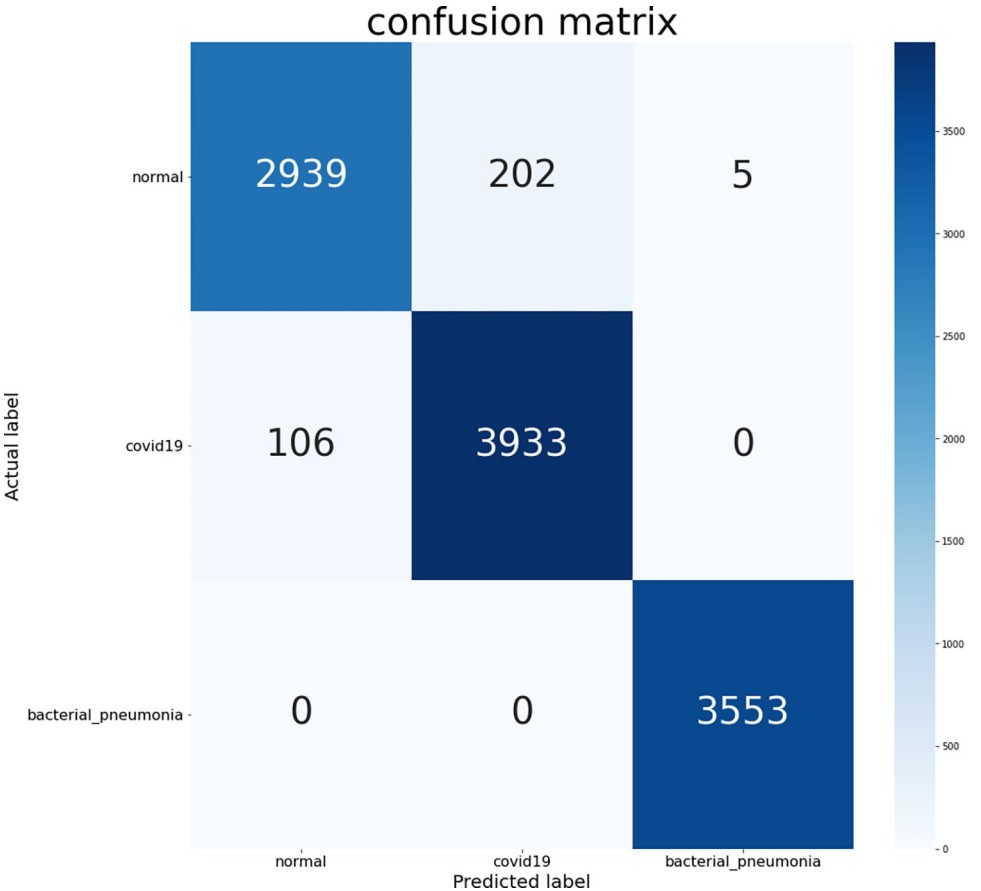

**Fig 8. The confusion matrix of the proposed model.**

dataset. To verify the performance of the proposed model, this study also explores the fusion effect of feature maps with different sizes of convolutional kernels and global information

From Table 6, we can see that the Feature extraction sub-network achieves better results when the convolutional kernel size is 5. Compared with other convolutional kernel size settings, this sub-network can extract more accurate detailed features and can better complement the local information missing from the global features, thus achieving better results.

### Mutual fusion and one-way fusion

Our network demonstrates that the fusion of global feature and local feature gets excellent results in Covid-19 classification. The comparations include the fusion from CNN to

**Table 3. Performance comparison of the proposed model with some DL-based models on the present dataset.**

| Model | Accuracy | Precision | Recall | F1 score |
|---|---|---|---|---|
| Vgg16 [16] | 0.9558 | 0.9560 | 0.9544 | 0.9551 |
| ResNet-18 [18] | 0.9363 | 0.9503 | 0.9478 | 0.9490 |
| ResNet-50 [18] | 0.9565 | 0.9561 | 0.9558 | 0.9559 |
| DenseNet121 [19] | 0.9584 | 0.9583 | 0.9579 | 0.9584 |
| ViT [35] | 0.9175 | 0.9172 | 0.9153 | 0.9161 |
| **Proposed model** | **0.9709** | **0.9716** | **0.9693** | **0.9704** |

**Table 4. Performance comparison of the proposed model with some other models regarding the detection of Covid-19.**

| Model | Accuracy | Precision | Recall | F1 score |
|---|---|---|---|---|
| Civit-Masot [22] | 0.86 | 0.86 | 0.86 | 0.86 |
| Ozturk [23] | 0.8702 | 0.8996 | 0.8535 | 0.8737 |
| Apostolopoulos [24] | 0.9348 | - | 0.9285 | - |
| Yoo [25] | 0.95 | 0.94 | **0.97** | - |
| Sethy [26] | - | - | 0.9533 | 0.9534 |
| **Proposed model** | **0.9709** | **0.9716** | 0.9693 | **0.9704** |

**Table 5. Performance comparison of the proposed model with possible module combinations.**

| Possible module combinations | Accuracy | | |
|---|---|---|---|
| | Transformer_block1 | Transformer_block2 | Transformer_block3 |
| Conv_block1 | **0.9709** | 0.9565 | 0.9486 |
| Conv_block2 | 0.9608 | 0.9556 | 0.9384 |
| Conv_block3 | 0.9497 | 0.9336 | 0.9219 |

**Table 6. Performance comparison of the proposed model with different convolution kernel sizes.**

| Model | Accuracy | Precision | Recall | F1 score |
|---|---|---|---|---|
| Conv3 | 0.9556 | 0.9557 | 0.9542 | 0.9549 |
| Conv5 | **0.9709** | **0.9716** | **0.9693** | **0.9704** |
| Conv7 | 0.9506 | 0.9502 | 0.9498 | 0.9500 |

**Table 7. Performance comparison of the proposed model with different fusion methods.**

| Fusion method | Accuracy | Precision | Recall | F1 score |
|---|---|---|---|---|
| The fusion from CNN to transformer | 0.9492 | 0.9500 | 0.9472 | 0.9484 |
| The fusion form transformer to CNN | 0.9509 | 0.9502 | 0.9506 | 0.9504 |
| **The mutual fusion** | **0.9709** | **0.9716** | **0.9693** | **0.9704** |

transformer, from transformer to CNN, and the mutual fusion of CNN and transformer. The fusion from CNN to transformer is to fuse the local features extracted by CNN and feed them to transformer, but the global features extracted by transformer are not fused with the CNN features; The fusion from transformer to CNN is to fuse the global features from the transformer feed them to CNN, but the local features from the CNN do not fuse with the transformer features; The mutual fusion of CNN and transformer is that the local features are fused with the global features and they are fed into the transformer and CNN. Table 7 compares the effect of three fusion methods and the results show that mutual fusion works best.

## 5. Conclusions

In this study, a CNN-transformer fusion network is proposed for Covid-19 image classification. This network could make the best of the different feature extraction capabilities of the CNN and the transformer, the CNN module and the transformer module are able to extract the local features and the global features of medical images, respectively. Therefore, this network fuses local lesion features and scattered global features to achieve classification, focusing

on both the global and the local features. The experiments show that the proposed network performs better than other DL-based models for the classification of Covid-19, bacterial pneumonia, and the normal. The comparison of the proposed model with other models concerning Covid-19 reveals that our model is good at detecting Covid-19 in CXR images, and achieves superior results compared to other models. There is still room for improvement in the following areas. Decentralized data from different institutions is required to improve the classification of the model. The model remains to be refined to distinguish between more types of pneumonia disease, and further developed as a computer-aided diagnosis system for pneumonia.

## Author Contributions

**Conceptualization:** Kai Cao, Tao Deng.

**Data curation:** Chuanlin Zhang.

**Funding acquisition:** Tao Deng.

**Investigation:** Kai Cao.

**Methodology:** Kai Cao.

**Project administration:** Tao Deng.

**Supervision:** Tao Deng.

**Validation:** Lin Li.

**Visualization:** Chuanlin Zhang, Limeng Lu.

**Writing – original draft:** Kai Cao.

**Writing – review & editing:** Kai Cao, Tao Deng.

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
