## [Decision Letter · Decision Letter 0]

26 Sep 2022

PONE-D-22-24175A CNN-transformer fusion network for COVID-19 CXR image classificationPLOS ONE

Dear Dr. Deng,

Thank you for submitting your manuscript to PLOS ONE. After careful consideration, we feel that it has merit but does not fully meet PLOS ONE’s publication criteria as it currently stands. Therefore, we invite you to submit a revised version of the manuscript that addresses the points raised during the review process.

Specifically, the authors need to improve the description of methodology and compare the results with the state-of-the-art.

We look forward to receiving your revised manuscript.

Kind regards,

Robertas Damaševičius

Academic Editor

PLOS ONE

Journal Requirements:

2. Please note that PLOS ONE has specific guidelines on code sharing for submissions in which author-generated code underpins the findings in the manuscript. In these cases, all author-generated code must be made available without restrictions upon publication of the work. Please review our guidelines at https://journals.plos.org/plosone/s/materials-and-software-sharing#loc-sharing-code and ensure that your code is shared in a way that follows best practice and facilitates reproducibility and reuse. New software must comply with the Open Source Definition.

Additional Editor Comments:

The authors need to improve the description of methodology and compare the results with the state-of-the-art.

Reviewers' comments:

Reviewer's Responses to Questions

**Comments to the Author**

1. Is the manuscript technically sound, and do the data support the conclusions?

Reviewer #1: Yes

Reviewer #2: Yes

2. Has the statistical analysis been performed appropriately and rigorously? 

Reviewer #1: Yes

Reviewer #2: I Don't Know

3. Have the authors made all data underlying the findings in their manuscript fully available?

Reviewer #1: Yes

Reviewer #2: Yes

4. Is the manuscript presented in an intelligible fashion and written in standard English?

Reviewer #1: Yes

Reviewer #2: No

5. Review Comments to the Author

Reviewer #1: The proposed machine learning solution is interesting and worth investigating. Moreover, comparison with state of art shows great efficiency. However, i cannot agree that the paper is ready for publication because of:

1) Discuss your proposition according to the newest approaches, especially in terms of other methods than a neural network. Why the proposed approach is worthy of investigation? For easier analysis, see such ideas as lung segmentation by a heuristic fox.

2) How did you model the architecture of CNN?

3) Some comparisons with learning transfer should be made.

4) Discussion on results is very limited.

Reviewer #2: Abstract: The global health crisis due to the fast spread of coronavirus disease (Covid-19) has

caused great danger to all aspects of healthcare, economy and other aspects. The

highly infectious and insidious nature of the new coronavirus greatly increases the

difficulty of outbreak prevention and control. The early and rapid detection of

Covid-19 is an effective way to reduce the spread of Covid-19. However, detecting the

Covid-19 accurately and quickly in large population remains to be a major challenge

worldwide. In this study, A CNN-transformer fusion framework is proposed for the

automatic classification of pneumonia on chest X-ray. This framework includes two

parts: data processing and image classification. The data processing stage is to

eliminate the differences between data from different medical institutions so that they

have the same storage format; in the image classification stage, we use a multi-branch

network with a custom convolution module and a transformer module, including

feature extraction, feature focus and feature classification sub-networks. Feature

extraction subnetworks extract the shallow features of the image and interact with the

information through the convolution and transformer modules. Both the local and

global features are extracted by the convolution module and transformer module of

feature focus subnetworks, and are classified by the feature classification subnetworks.

The proposed network could decide whether or not a patient has pneumonia,

furthermore differentiate between Covid-19 and bacterial pneumonia. This network

was implemented on the collected benchmark datasets and the result shows that

accuracy, precision, recall and F1 score are 97.09%, 97.16%, 96.93% and 97.04%,

respectively. Our network was compared with other researchers’ proposed methods

and achieved better results in terms of accuracy, precision and F1 score, proving that

it is superior for Covid-19 detection. With further improvements to this network, we

hope that it will provide doctors with an effective tool for diagnosing Covid-19.

Overall, the novelty is sufficient but still several issues remaining that should be incorporated.

1) "Covid-19 is a pulmonary disease caused by severe acute respiratory syndrome

coronavirus 2 (SARS-CoV-2) in 2019 and is highly infectious, mutagenic, and Covid-19

and its novel mutant strains such as delta, omicron, and omicron XE variant have"- add the following two works for the support of this statement (A Healthcare System for COVID19 Classification Using Multi-Type Classical Features Selection; A Rapid Artificial Intelligence-based Computer-Aided Diagnosis System for COVID19 Classification from CT Images )

2) Add the following studies in the related work, as the related work is not suffiecint and up to date.

- COVID19 Classification using Chest X-Ray Images: A Framework of CNN-LSTM and Improved Max Value Moth Flame Optimization

- COVID-19 Classification from Chest X-Ray Images: A Framework of Deep Explainable Artificial Intelligence

3) "According to the WHO, as of April 4, 2022, there were 489,779,062 confirmed cases of COVID-19

including 6,152,095 deaths "- this statement need to be confirmed with correct data

4) "There are a number of reasons for the lower sensitivity, such as the possible inadequate supply of reagents, the lack of expertise required for the testing, low viral load in patients and the long testing cycles."- this statement need some justifications

5) "Medical imaging and deep learning (DL) can play an important role in

pre-detection efforts to combat disease. In recent years, researchers have used deep

neural networks to achieve remarkable results in a variety of fields. Recent advances in

DL show that computers can extract more information from images, more reliably,

and more accurately than ever before"- add the following works for the support of this important statement:

- Brain MRI analysis using deep neural network for medical of internet things applications

- BrainNet: optimal deep learning feature fusion for brain tumor classification

6) The major contributions should be revised, as the last contribution (3) is not a contribution.

7) Draw figure 1 in more attractive.

8) It is important to add the images before and after the data augmentationn.

9) Remove Eq. 6-9, these are well known.

10) In table 7, what is reason behind the choice of these hyperparameters.

6. PLOS authors have the option to publish the peer review history of their article (what does this mean?). If published, this will include your full peer review and any attached files.

Reviewer #1: No

Reviewer #2: No

---

## [Author Response · Author response to Decision Letter 0]

11 Oct 2022

The response has been included in submission system.

---

## [Decision Letter · Decision Letter 1]

13 Oct 2022

A CNN-transformer fusion network for COVID-19 CXR image classification

PONE-D-22-24175R1

Dear Dr. Deng,

We’re pleased to inform you that your manuscript has been judged scientifically suitable for publication and will be formally accepted for publication once it meets all outstanding technical requirements.

Kind regards,

Robertas Damaševičius

Academic Editor

PLOS ONE

Additional Editor Comments (optional):

Reviewers' comments:

Reviewer's Responses to Questions

**Comments to the Author**

1. If the authors have adequately addressed your comments raised in a previous round of review and you feel that this manuscript is now acceptable for publication, you may indicate that here to bypass the “Comments to the Author” section, enter your conflict of interest statement in the “Confidential to Editor” section, and submit your "Accept" recommendation.

Reviewer #1: All comments have been addressed

Reviewer #2: (No Response)

2. Is the manuscript technically sound, and do the data support the conclusions?

Reviewer #1: Yes

Reviewer #2: (No Response)

3. Has the statistical analysis been performed appropriately and rigorously? 

Reviewer #1: Yes

Reviewer #2: (No Response)

4. Have the authors made all data underlying the findings in their manuscript fully available?

Reviewer #1: Yes

Reviewer #2: (No Response)

5. Is the manuscript presented in an intelligible fashion and written in standard English?

Reviewer #1: Yes

Reviewer #2: (No Response)

6. Review Comments to the Author

Reviewer #1: All my comments have been addressed. In my opinion, the paper is ready for publication and can be accepted.

Reviewer #2: Authors well revised this manuscript as per my recommendation. I have no more comments and this paper is ready for publication.

7. PLOS authors have the option to publish the peer review history of their article (what does this mean?). If published, this will include your full peer review and any attached files.

Reviewer #1: No

Reviewer #2: No

---

## [Editor Report · Acceptance letter]

18 Oct 2022

PONE-D-22-24175R1 

A CNN-transformer fusion network for COVID-19 CXR image classification 

Dear Dr. Deng:

I'm pleased to inform you that your manuscript has been deemed suitable for publication in PLOS ONE. Congratulations! Your manuscript is now with our production department. 

Kind regards, 

on behalf of

Professor Robertas Damaševičius 

Academic Editor

PLOS ONE